# Buckwheat Flour and Its Starch Prevent Age-Related Cognitive Decline by Increasing Hippocampal BDNF Production in Senescence-Accelerated Mouse Prone 8 Mice

**DOI:** 10.3390/nu14132708

**Published:** 2022-06-29

**Authors:** Shigeru Katayama, Chizuru Okahata, Masashi Onozato, Takaaki Minami, Masanaga Maeshima, Kazuaki Ogihara, Shinya Yamazaki, Yuta Takahashi, Soichiro Nakamura

**Affiliations:** 1Department of Agriculture, Graduate School of Science and Technology, Shinshu University, 8304 Minamiminowa, Kamiina, Nagano 399-4598, Japan; 20as202k@shinshu-u.ac.jp (C.O.); snakamu@shinshu-u.ac.jp (S.N.); 2Department of Biomolecular Innovation, Institute for Biomedical Sciences, Interdisciplinary Cluster for Cutting Edge Research, Shinshu University, 8304 Minamiminowa, Kamiina, Nagano 399-4598, Japan; 3Nikkoku Seifun Co., Ltd., 1-16-2 Minamichitose, Nagano 380-0823, Japan; m.kaihatu1@nikkoku.co.jp (M.O.); minamit@nikkoku.co.jp (T.M.); maeshimam@nikkoku.co.jp (M.M.); 4Nagano Prefecture General Industrial Technology Center, 205-1 Kurita, Nagano 380-0921, Japan; ogihara-kazuaki-r@pref.nagano.lg.jp (K.O.); yamazaki-shinya-r@pref.nagano.lg.jp (S.Y.); takahashi-yuta-r@pref.nagano.lg.jp (Y.T.)

**Keywords:** buckwheat, brain-derived neurotrophic factor, cognitive decline, senescence-accelerated mouse prone 8, starch

## Abstract

Buckwheat is an important pseudo-cereal crop worldwide. This study investigated whether long-term administration of buckwheat can suppress age-related cognitive decline in senescence-accelerated mouse prone 8 (SAMP8) mice. For 26 weeks, 18-week-old male SAMP8 mice were fed a standard diet containing 5% (*w*/*w*) buckwheat, Tartary buckwheat, wheat, or rice flour. In the Barnes maze and passive avoidance tests, mice fed buckwheat whole flour (BWF) showed improved cognitive performance compared to those fed a control diet, while no improvement was noticed in case of the other diets. Analysis of the gut microbiota showed that BWF and buckwheat outer flour administration increased the abundance of *Lactococcus* and *Ruminiclostridium*, respectively, at the genus level. The expression levels of brain-derived neurotrophic factor (BDNF), postsynaptic Arc and PSD95, and the mature neuronal marker NeuN in the hippocampus were increased after BWF administration, which was induced by the activation of the ERK/CREB signaling pathway and histone H3 acetylation. A similar increase in cognitive performance-related hippocampal BDNF expression in SAMP8 mice was observed after the oral administration of starch prepared from BWF. Therefore, the long-term administration of BWF suppresses cognitive decline by increasing hippocampal BDNF production in SAMP8 mice.

## 1. Introduction

Aging is an inevitable part of human life and is associated with a decline in the physical and functional capacities of tissues and organs. Major debilitating illnesses related to aging include mild cognitive impairment and dementia, including Alzheimer’s disease. Physical exercise, dietary interventions, and the daily intake of vegetables and fruits rich in various phytochemicals contribute to maintaining good health and quality of life during the aging process. The interest in dietary approaches for the prevention of cognitive impairment and dementia has been increasing.

*Fagopyrum esculentum* Moench (buckwheat) is a highly nutritious pseudo-cereal that contains large amounts of starch, protein, dietary fiber, phenolic compounds, and minerals [1,2]. Buckwheat is used to prepare noodles, pasta, blended bread, and other flour-based products, especially Japanese buckwheat noodles (soba noodles), and is consumed as a gluten-free functional food by people with celiac disease [3]. The taste, color, and chemical composition of buckwheat products depend on the species and groat portion [4,5,6]. Two major varieties of agricultural significance are common buckwheat (*F. esculentum* Moench) and Tartary buckwheat (*Fagopyrum tataricum* L. Gaertn.), which are known as sweet buckwheat and bitter buckwheat, respectively. Tartary buckwheat contains higher concentrations of bitter phenolic compounds, particularly rutin, than those in common buckwheat [7]. The contents of protein and ash in buckwheat flour fractions increase from the inner to outer fractions [5]. The inner fractions, containing mostly starch, are considerably whiter than the outer fractions, which contain higher amounts of protein and dietary fiber [4]. Therefore, different buckwheat species and fractionated flours may contribute not only to the different qualities of buckwheat products, such as noodles, but also to different health benefits.

Certain bioactive food components potentially prevent age-related cognitive decline in different animal models. Senescence-accelerated mouse (SAM) strains have been established as models of accelerated aging, and the strain SAM prone 8 (SAMP8) spontaneously exhibits age-related behavioral disorders, including shortened life span and cognitive impairment [8,9]. Our studies using SAMP8 mice have indicated that prolonged intake of a diet supplemented with soybean peptide and soymilk residue (okara) suppresses age-related cognitive decline compared to that in controls, which is mediated by significant upregulation of brain-derived neurotrophic factor (BDNF) expression via activation of cAMP regulatory element binding protein (CREB) transcription in the brain [10,11]. Since initial cognitive decline is generally considered to result from dysfunctional synaptic plasticity, the induction of neuroprotective proteins, including neurotrophic factors, protein chaperones, and antioxidant enzymes, by dietary bioactive compounds appears to contribute to the prevention of age-related cognitive decline [12,13,14].

The aims of this study were to determine the effects of buckwheat flour on age-related cognitive decline in SAMP8 mice and explore the underlying mechanisms. We then compared the effects of different fractionated portions of buckwheat flour and different species. We found that long-term administration of buckwheat whole flour (BWF) suppresses cognitive decline by increasing hippocampal BDNF production in SAMP8 mice.

## 2. Materials and Methods

### 2.1. Materials

BWF, buckwheat outer flour (BOF), Tartary buckwheat whole flour (TWF), wheat flour (WF), rice flour (RF), buckwheat starch (BS), wheat starch (WS), and rice starch (RS) were obtained from Nikkoku Seifun Co. Ltd. (Nagano, Japan). The chemical compositions of the buckwheat, rice, and wheat flour are summarized in Table 1.

### 2.2. Animals

Male SAMP8 and SAM resistant 1 (SAMR1) mice were purchased from Japan SLC Inc. (Shizuoka, Japan). All mice were housed individually (one per cage) to prevent them from fighting, and the animal room was maintained under controlled temperature (20−23 °C) and humidity (40−70%) with a 12/12 h light/dark cycle (lights on at 8:00 a.m.). All animal experiments were performed in accordance with the animal experimental protocol approved by the Institutional Animal Care and Use Committee of Shinshu University (permission No. 019011).

### 2.3. Animal Treatment

#### 2.3.1. Study 1

Eighteen-week-old SAMP8 mice were divided into seven groups: standard diet–fed group (AIN-93M, Oriental Yeast Co. Ltd., Tokyo, Japan) (control), 5% (*w*/*w*) BWF-, BOF-, TWF-, WF-, and RF-containing diet–fed groups, and 0.05% (*w*/*w*) rutin-containing diet–fed group (*n* = 10 in each group), as shown in Table 2. The contents of protein, lipid, carbohydrate, and dietary fiber provided by BWF, BOF, TWF, WF, or RF were calculated to balance the ingredients provided in AIN-93M diet. SAMR1 mice (*n* = 8) were used as the normal aging control and fed the AIN-93M diet. Mice were allowed free access to food and tap water. Food intake and body weight were recorded every week. At 42–43 weeks of age, all mice were subjected to the Barnes maze and passive avoidance tests to assess their cognitive performance. All mice were euthanized at 44 weeks of age after the last memory test, and the brain and cecal contents were collected. For Western blotting, the hippocampal tissues of mice brain were frozen in liquid nitrogen. For immunostaining, the brain tissues were fixed in paraformaldehyde and subsequently embedded in paraffin.

#### 2.3.2. Study 2

Eighteen-week-old SAMP8 mice were divided into five groups: standard diet–fed group (AIN-93M) (control), and 5% (*w*/*w*) BWF- or 4% (*w*/*w*) BS-, WS-, or RS-containing diet–fed groups (*n* = 10 in each group), as shown in Table 3. The contents of protein, lipid, carbohydrate, and dietary fiber provided by BWF, BS, WS, or RS were calculated to balance the ingredients in AIN-93M diet. SAMR1 mice (*n* = 8) fed AIN-93M diet were used as normal aging control. Mice were allowed free access to food and tap water. Food intake and body weight were recorded every week. At 26–27 weeks of age, all mice were subjected to the Barnes maze and passive avoidance tests. In study 2, treatment was finished when the mice were 28 weeks of age because SAMP8 mice begin to die at 30 weeks of age. All mice were euthanized, and the hippocampus was collected, frozen in liquid nitrogen, and stored at −80 °C until analysis.

### 2.4. Barnes Maze Test

The Barnes maze test was performed as previously described [15]. The maze consisted of a grey circular platform (90 cm in diameter) with 20 equally spaced holes (5 cm in diameter) located 3 cm away from the perimeter (Muromachi Kikai Co. Ltd., Tokyo, Japan). A black escape box was placed under one of the holes. This circular platform was mounted on top of a steel stool, which was 90 cm above the ground and balanced. Visual cues were placed on the walls (triangles and squares) of the experimental room. The maze was divided into four quadrants (45°, 90°, 135°, and 180°) in clockwise and counterclockwise directions from the escape box (0°). Each animal was subjected to 1 d of habituation and 4 d of training comprising three trials per day. During training sessions, the mouse was placed in the middle of the maze for 10 s and then allowed to freely explore the platform until either it entered the escape box or 3 min had elapsed. If the mouse did not find the escape box within 3 min, it was placed within the escape box. The next day after acquisition training, mice went through a memory probe trial. The escape box was removed during this trial, and mice were placed in the center of the maze for 10 s. Each mouse was given 3 min to explore the maze and search for the escape box. The escape latency (time taken to enter the escape box) and the percentage of time spent in the target quadrant (geometrical area covering 25% of the Barnes maze with target hole) were recorded using an Any-maze video tracking system (Stoelting Co., Wood Dale, IL, USA).

### 2.5. Passive Avoidance Test

The passive avoidance test was performed using a step-through test cage (Muromachi Kikai Co. Ltd., Tokyo, Japan) consisting of white and black compartments separated by a sliding door. During the training phase, each mouse was placed in the light compartment and allowed to explore for 10 s. The door was opened and the step-through latency was recorded. After the mice entered the dark compartment, the door was immediately closed and a mild foot shock of 0.2 mA was applied for 3 s. Training sessions were conducted for two consecutive days. The probe test was performed using the same procedure without any shocks. The step-through latency to enter the dark compartment was recorded. A maximum retention latency of 300 s was allowed for mice that did not enter the dark compartment.

### 2.6. Western Blot Analysis

Hippocampal tissues were lysed in radioimmunoprecipitation assay lysis buffer (Santa Cruz Biotechnology, Santa Cruz, CA, USA), and proteins were extracted for Western blot analysis. Protein content in the lysates was measured using the bicinchoninic acid assay (Fujifilm Wako Pure Chemical, Osaka, Japan). The protein solution (4 mg) was separated using 10% sodium dodecyl sulfate-polyacrylamide gel electrophoresis and transferred to polyvinylidene fluoride membranes (Merck Millipore, Burlington, MA, USA). The membranes were subsequently blocked with Blocking One and Blocking One-P (Nacalai Tesque, Kyoto, Japan) for 20 min at room temperature and then incubated with primary antibodies against neuronal nuclear antigen (NeuN), activity-regulated cytoskeleton-associated protein (Arc), BDNF, β-actin (Abcam, Cambridge, United Kingdom), postsynaptic density protein 95 (PSD95) (Gene Tex, Irvine, CA, USA), CREB, phosphorylated (p)-CREB (Sigma–Aldrich, St. Louis, MA, USA), calcium/calmodulin-dependent protein kinase II (CaMKII), p-CaMKII, extracellular signal-regulated kinase (ERK) 1/2, p-ERK 1/2, histone H3, and acetyl-histone H3 (Cell Signaling Technology, Danvers, MA, USA) overnight at 4 °C. After washing, the membranes were incubated with a secondary antibody, horseradish peroxidase–conjugated goat anti-rabbit IgG or goat anti-mouse IgG (Santa Cruz Biotechnology) for 1 h at room temperature. For detection, luminescence reaction was performed using a luminescent substrate solution (ATTO, Tokyo, Japan) and photographs were obtained using a chemiluminescent photographing apparatus Ez Capture MG (AE-9300H-CS, ATTO). The target proteins were analyzed using Image Saver 5 analysis software (ATTO).

### 2.7. Immunostaining

Paraffin-embedded mouse brain sections were dewaxed using xylene and hydrated in decreasing concentrations of ethanol. Sections were boiled in 10 mM Tris/1 mM EDTA buffer (pH 9.0) for 20 min and cooled for 30 min at room temperature for antigen retrieval. The sections were then washed twice with Tris-buffered saline (TBS). After 1 h of blocking with 5% bovine serum albumin in TBS, the sections were incubated with an antibody against BDNF (Abcam) overnight at 4 °C. The slides were washed twice in TBS and incubated with the secondary antibody, Alexa Fluor 488 goat anti-rabbit IgG (H&L) (Abcam). After washing twice with TBS, the sections were mounted with ImmunoSelect anti-fading mounting medium DAPI (Dianova, Hamburg, Germany) and examined under a fluorescence microscope (EVOS fl; Advanced Microscopy Group, Bothell, WA, USA).

### 2.8. Microbiological Analysis

Five mice were randomly chosen from each group, and genomic DNA (gDNA) was extracted from the cecal contents using the NucleoSpin DNA Stool Extraction Kit (Takara Bio, Shiga, Japan). The purified gDNA samples were sent to GeneWiz (South Plainfield, NJ, USA) for 16S rRNA gene sequencing analysis for microbiota profiling with barcoded amplicons of the V3-V4 regions. Sequencing results were analyzed using QIIME 1.9.1, and taxonomic assignment of operational taxonomic units (OTUs) was performed using the Silva v132 database. Analysis of alpha and beta diversities, principal coordinate analysis (PCoA), principal component analysis (PCA), and linear discriminant analysis effect size (LEfSe) were performed to determine the characteristics of the gut microflora of different groups.

### 2.9. Statistical Analyses

Results are expressed as the mean ± standard error of the mean. The data of training sessions in the Barnes maze were analyzed using the two-way repeated measures analysis of variance (ANOVA), and other data were analyzed by one-way ANOVA, followed by Dunnett’s post hoc test (GraphPad Prism 5.03; GraphPad Software, San Diego, CA, USA). A *p* value < 0.05 was considered significant. All data are expressed as the mean ± standard error of the mean.

## 3. Results

### 3.1. Effects of Buckwheat Flour on the Cognitive Function of SAMP8

We first investigated whether long-term administration of buckwheat flour could suppress age-related cognitive decline in SAMP8 mice. No significant changes in the body weight and feed consumption of the mice were observed throughout the experiment (data not shown). One SAMP8 mouse administered BOF and WF died at 35 and 34 weeks of age, respectively. During the training sessions with the Barnes maze, SAMP8 mice administered BWF (P8-BWF) and rutin found the target hole earlier than did the SAMP8 control (P8-Ct) mice on the fourth day (*p* < 0.05), while SAMP8 mice administered BOF (P8-BOF) found hole earlier than did P8-Ct on the second day (*p* < 0.05) (Figure 1A). P8-BWF found the target hole earlier than did P8-Ct in the probe trial (Figure 1B). In addition, P8-BWF spent significantly more time in the target quadrant during the probe trial (Figure 1C). No significant changes were observed after treatment with BOF, TWF, rutin, WF, or RF. No statistical differences were observed in escape latency and time in quadrant of the Barnes maze test between the SAMR1 control (R1-Ct) and P8-Ct groups; however, a significant difference in step-through latency of the passive avoidance test was observed between the R1-Ct and P8-Ct groups (*p* < 0.05) (Figure 1D). BWF treatment appeared to increase the latency compared to that of the P8-Ct group, although no significant differences were found.

### 3.2. Effects of Buckwheat Flour on the Gut Microbiota Community of SAMP8

The α- and β-diversity indices of the gut microflora in mice subjected to different interventions were investigated to determine the effect of the buckwheat flour treatment on the gut microbiome. There was no significant difference in the α-diversity of both Chao1 and Shannon indices in the P8-Ct group compared to that in the R1-Ct group (Figure 2A,B). In contrast, oral treatment with BWF increased the Chao1 (*p* < 0.05) and Shannon (*p* = 0.068) indices compared with those in the P8-Ct group. BOF treatment significantly increased the Chao1 (*p* < 0.01) and Shannon (*p* < 0.05) indices compared with those in the P8-Ct group. We also assessed the dissimilarity in individual microbial communities (β-diversity). The microbial communities were separated via PCA, although there was no distinct difference in the PCoA (Figure 2C,D). LEfSe analysis was performed to further identify specific individual taxa that were differentially enriched at the class to species level among the groups. Bacteria enriched in the P8-BWF group belonged to the family *Muribaculaceae* (Figure 2E). Bacteria from the class *Bacilli*, the order *Lactobacillades*, the genus *Anaerosipes* were enriched in the P8-BOF group. Furthermore, according to the taxonomic classification analysis of OTUs, the genus *Lactococcus* was enriched after treatment with BWF (*p* < 0.05), and *Ruminiclostridium* was enriched after treatment with BOF compared with that in the P8-Ct group (*p* < 0.01) (Figure 2F,G).

### 3.3. Regulatory Effects of Buckwheat Flour on Neurons and Neuroplasticity-Related Proteins

Western blot analysis indicated that the hippocampal levels of the mature neuronal marker NeuN and the postsynaptic marker PSD95 increased after oral administration of BWF in mice compared to those in the P8-Ct and P8-BOF groups (*p* < 0.05) (Figure 3A,B). Similar increases in the levels of neuroplasticity markers BDNF and Arc were observed when mice were orally administered BWF (*p* < 0.05). In contrast, no increase was observed in the P8-BOF group. Furthermore, immunohistochemical staining showed that the P8-Ct group had significantly fewer BDNF-positive cells in the hippocampus than the R1-Ct group (*p* < 0.05), and BWF treatment increased the number of BDNF-positive cells compared to that in the P8-Ct group (*p* < 0.05) (Figure 4A,B). We next investigated the phosphorylation levels of key signaling molecules, such as ERK1/2, CaMKII, and CREB, in the hippocampus. BWF treatment induced a significant increase in the phosphorylation of ERK (*p* < 0.05) and CREB (*p* < 0.01) compared to that in P8-Ct mice; however, the level of phosphorylated CaMKII remained unchanged in the P8-BWF group (Figure 5A,B). Furthermore, a significant increase in acetyl-H3 was observed in the P8-BWF group (*p* < 0.01). No significant changes were observed in the P8-BOF group.

### 3.4. Effects of Buckwheat Starch on the Cognitive Function and Hippocampal BDNF Expression in SAMP8 Mice

We further investigated whether long-term administration of BS could suppress age-related cognitive decline in SAMP8 mice. Throughout the experiment, no significant changes in the body weight and feed consumption of mice were observed (data not shown), and no mice died. During the training sessions of the Barnes maze, P8-BWF and SAMP8 mice administered BS (P8-BS) found the target hole earlier than did P8-Ct on the fourth day (*p* < 0.05) (Figure 6A). In the probe trials, both P8-BWF and P8-BS found the target hole earlier than did P8-Ct (Figure 6B). The time spent in the target quadrant during the probe trial remained unchanged after oral administration of BWF and BS (Figure 6C). In the passive avoidance test, the latency into the dark chamber of P8-Ct was shorter than that of R1-Ct (*p* < 0.01). Treatment with BWF or BS effectively extended the latency compared to that of the P8-Ct group (*p* < 0.05) (Figure 6D). Furthermore, a significant increase in hippocampal BDNF expression was observed after oral administration of BS (*p* < 0.05) (Figure 6E).

## 4. Discussion

The induction of neuroprotective proteins, including neurotrophic factors, protein chaperones, and antioxidant enzymes, by dietary bioactive compounds may contribute to the prevention of age-related cognitive decline. Therefore, we investigated the effects of buckwheat flour on age-related cognitive decline in SAMP8 mice and determined the underlying mechanisms. Treatment was initiated when the mice were 18 weeks of age, which is the pre-onset stage of cognitive dysfunction in this strain. The Barnes maze test was performed to evaluate spatial learning and memory, and then the passive avoidance test was performed to evaluate associative learning and memory. The present study showed that long-term administration of BWF attenuated age-related cognitive decline in SAMP8 mice. Oral treatment with WF or RF did not show any improvements in learning and memory. Rutin has been reported to exhibit neuroprotective activity as a potent antioxidant present in buckwheat [16]; however, treatment with rutin and TWF containing a high content of rutin showed no increased improvement in learning and memory compared to that of the P8-Ct.

To determine the mechanisms underlying the effects of oral administration of BWF, we focused on two molecules involved in learning and memory processes, BDNF and Arc [17,18]. BDNF is a crucial mediator of neuronal vitality and function and regulates morphology by acting on both presynaptic and postsynaptic processes. It is considered a major factor in neuronal events that underlie learning and memory. We found that hippocampal BDNF levels in the P8-Ct group were lower than those in the R1-Ct group, which is consistent with previous data on age-related decline in BDNF expression. Furthermore, compared to that in the P8-Ct group, BWF treatment increased hippocampal BDNF levels via the activation of the ERK/CREB signaling pathway. BDNF not only promotes neuronal survival and differentiation but also regulates synaptic transmission and plasticity in the central nervous system. An increase in hippocampal PSD95 expression level was observed in the P8-BWF group. PSD95 has various functions at the synapse and is implicated in the regulation of ion channel function, synaptic activity, and intracellular signaling [19]. Arc is a downstream target of BDNF and is deeply involved in synaptic plasticity as a postsynaptic protein [20]. Therefore, the increased PSD95 expression induced by BDNF may represent a potential molecular substrate by which BWF increases hippocampal synaptic plasticity. In addition, BWF treatment increased NeuN expression level in the hippocampus. The increase in NeuN expression, which correlates with the increase in BDNF level following BWF treatment, may prevent neuronal loss or induce the formation of new neuronal cells. Histone acetylation and deacetylation have been reported to be deeply involved in synaptic plasticity and neural function, and histone deacetylase inhibition enhances memory formation [21,22]. An increase in histone H3 acetylation in the hippocampus was observed in the P8-BWF group. These results suggest that BWF intake suppressed cognitive decline by enhancing synaptic plasticity in the central nervous system.

Dietary fiber has been studied for its impact on cognitive function and mental health by affecting gut–brain axis communication [23]. Since buckwheat grain contains relatively high amount of dietary fiber, we investigated the effects of BWF and BOF on the gut microbiota. A significant increase in *Lactococcus* was observed in the P8-BWF group. *Lactococcus* is a genus of Gram-positive lactic acid bacteria that are widely used as starting cultures in manufacturing fermented dairy products, such as cheese and yogurt [24]. Different species of this genus have been already explored for their potential beneficial role to the brain. *Lactococcus lactis* subsp. *cremoris* LL95 exerts antidepressant-like effects in lipopolysaccharide-induced depression in mice by reducing oxidative status and inflammatory responses in the hippocampus [25]. *L. lactis* strain WHH2078 can alleviate depressive- and anxiety-like behaviors in mice with induced chronic unpredictable mild stress by improving 5-hydroxytryptamine metabolism, restoring BDNF levels, and modulating the gut microbiome composition [26]. Furthermore, heat-killed *L. lactis* KC24 shows neuroprotective effects by increasing BDNF expression and decreasing apoptosis in oxidatively stressed SH-SY5Y cells [27]. Bacterial commensals in the gut communicate with the central nervous system and regulate the brain neurochemistry and behavior in several ways, including the production of bacterial metabolites and immune mediators and mediating direct brain signaling via the vagus nerve [28]. Further investigation is necessary to elucidate the potential benefits of *Lactococcus* on cognitive function.

Buckwheat contains high amount of buckwheat protein that is known to be digestion-resistant [29]. The outer fraction of buckwheat seeds is rich in low-digestible protein and dietary fiber. In this study, a significant difference in α-diversity, including Chao1 and Shannon indices, was observed between the P8-Ct and P8-BOF groups. In addition, *Ruminiclostridium* was enriched in the BOF group compared to that in the P8-Ct group. Despite these differences in the gut microbiota, oral treatment with BOF had no effects on learning and memory and hippocampal BDNF levels. *Ruminiclostridium* is a genus of anaerobic bacteria capable of depolymerizing cellulose [30] and producing short-chain fatty acids (SCFAs) such as acetic and butyric acids [31]. SCFAs can affect the function and development of the central nervous system [32]; however, no significant changes in SCFAs levels in the feces were observed between the P8-Ct and P8-BOF groups (data not shown). Therefore, the changes in the gut microbiota caused by BOF might not affect the cognitive performance of SAMP8 mice.

Since BWF has a high starch content, BS activity may have been closely related to the attenuation of cognitive decline in SAMP8 mice caused by BWF treatment. Long-term intake of BS showed a similar improvement in cognitive performance and hippocampal BDNF expression compared to that obtained using BWF, suggesting that BS can mainly function as an active compound mediating these effects. Starch is composed of amylose and amylopectin, and the average polymerization level of amylose in buckwheat is not considerably different from that in rice and wheat; however, the branched structure of amylopectin differs [33]. Future studies should identify the detailed characteristics of BS, including its branched structure.

In conclusion, our study demonstrated that long-term administration of BWF attenuated cognitive decline in a SAMP8 mouse model. Our findings suggest that prolonged consumption of not only the whole fraction of buckwheat flour but also its starch may induce BDNF expression in the hippocampus, contributing to sustained neuronal plasticity. Daily intake of BS may contribute to the prevention of age-related cognitive decline in elderly individuals.

## Figures and Tables

**Figure 1 nutrients-14-02708-f001:**
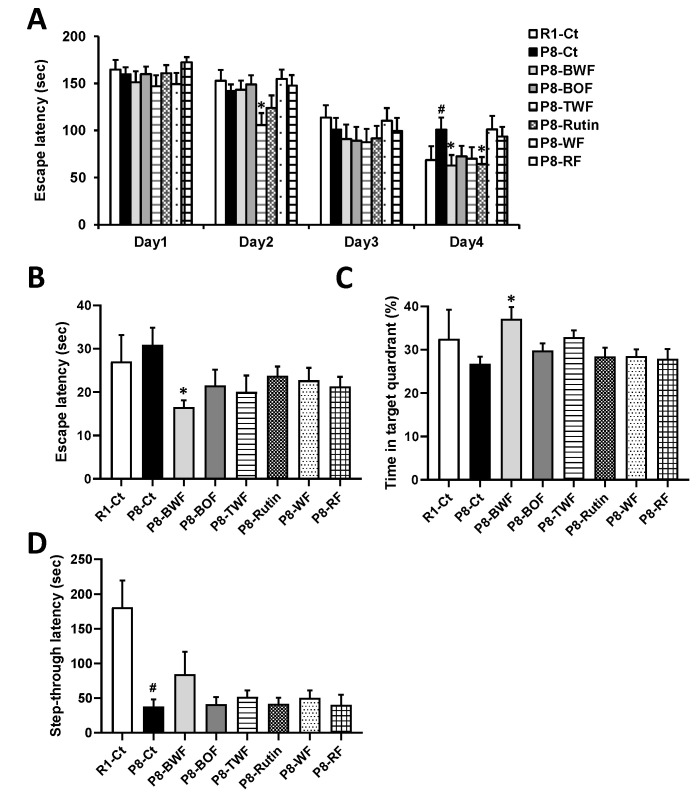
Effect of oral administration of various flours on learning and memory in SAMP8 mice. (**A**) Escape latency during training in the Barnes maze. (**B**) Escape latency during the probe test in the Barnes maze. (**C**) Time spent in target quadrant during the probe test in the Barnes maze. (**D**) Step-through latency in the passive avoidance test. Data are presented as the mean ± standard error of the mean (*n* = 8−10). ^#^
*p* < 0.05, compared with SAMR1 control (R1-Ct) group. * *p* < 0.05, compared with SAM8 control (P8-Ct) group. SAMP8, senescence-accelerated mouse prone 8; SAMR1, senescence-accelerated mouse resistant prone 1; BWF, buckwheat whole flour; BOF, buckwheat outer flour; TWF, Tartary buckwheat whole flour; WF, wheat flour; RF, rice flour.

**Figure 2 nutrients-14-02708-f002:**
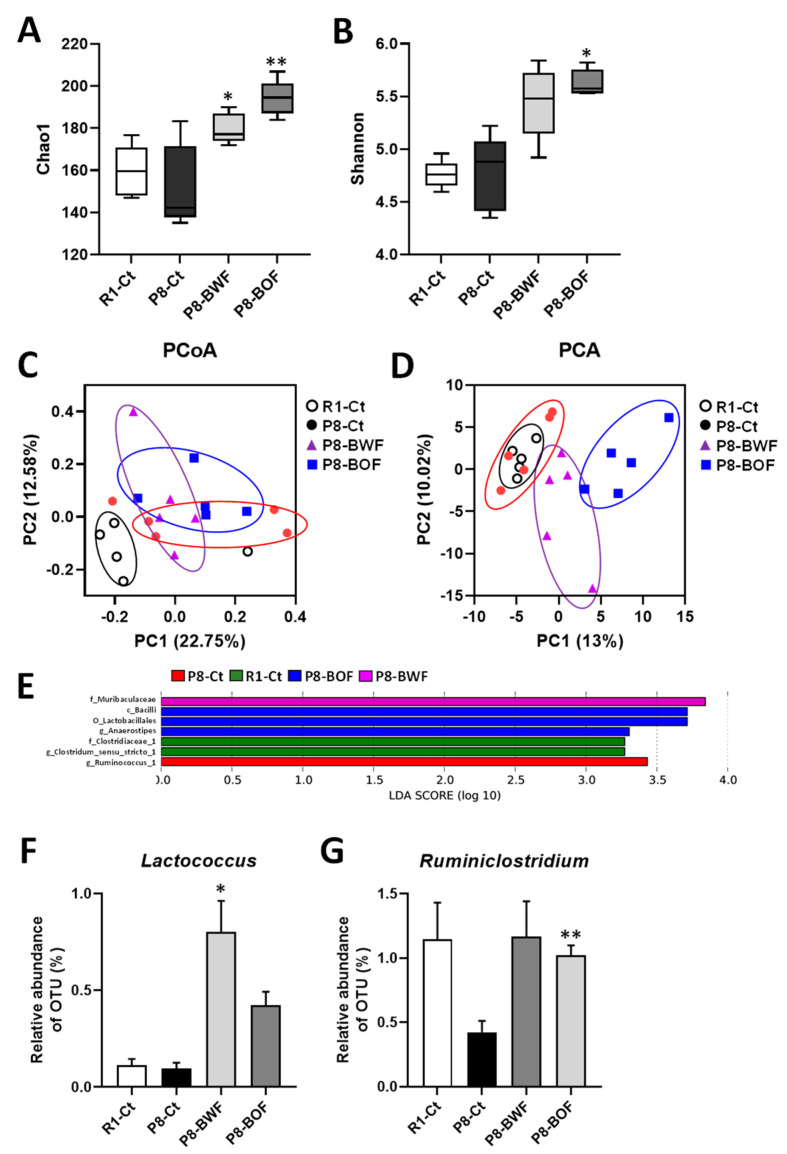
Effects of oral administration of buckwheat flours on the gut microbiota in SAMP8. (**A**) and (**B**) α-Diversity of flora in the cecal contents. (**C**,**D**) β-Diversity of flora in the cecal contents. (**E**) Results of the LEfSe analysis (LDA > 3.0). (**F**,**G**) Effects on *Lactococcus* and *Ruminiclostridium* abundances. Data are presented as the mean ± standard error of mean (*n* = 5). * *p* < 0.05, ** *p* < 0.01 vs. the P8-Ct group. LEfSe, linear discriminant analysis Effect Size; LDA, linear discriminant analysis; OTU, operational taxonomic unit.

**Figure 3 nutrients-14-02708-f003:**
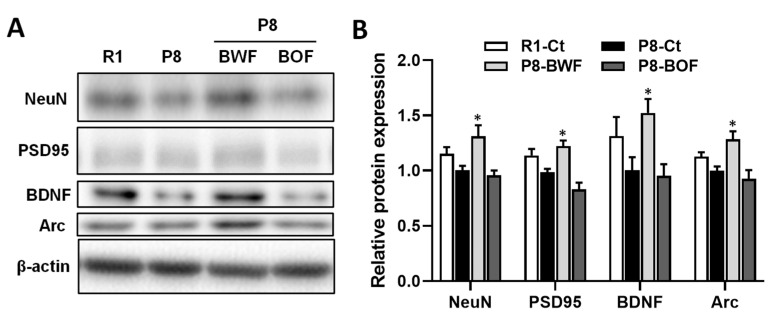
Effects of oral administration of buckwheat flours on protein expression levels in the hippocampus of SAMP8. (**A**) Representative image of the Western blot. β-actin was used as internal control. (**B**) Densitometric analysis of the band was expressed as relative fold compared with that in the P8-Ct group. Data are presented as the mean ± standard error of mean (*n* = 5). * *p* < 0.05, compared with the P8-Ct group. NeuN, neuronal nuclear antigen; PSD95, postsynaptic density protein 95; BDNF, brain-derived neurotrophic factor; Arc, activity-regulated cytoskeleton-associated protein.

**Figure 4 nutrients-14-02708-f004:**
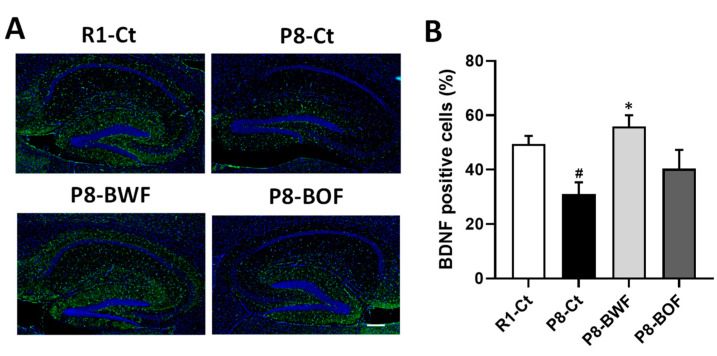
Effects of oral administration of buckwheat flours on BDNF distribution in the hippocampus of SAMP8. (**A**) Representative image of BDNF (green)-labeled brain section counterstained with 4′,6-diamidino-2-phenylindole (blue). Scale bar represents 300 μm. (**B**) Quantitative data of BDNF-positive cells. Data are presented as the mean ± standard error of mean (*n* = 5). ^#^
*p* < 0.05, compared with R1-Ct group. * *p* < 0.05, compared with the P8-Ct group.

**Figure 5 nutrients-14-02708-f005:**
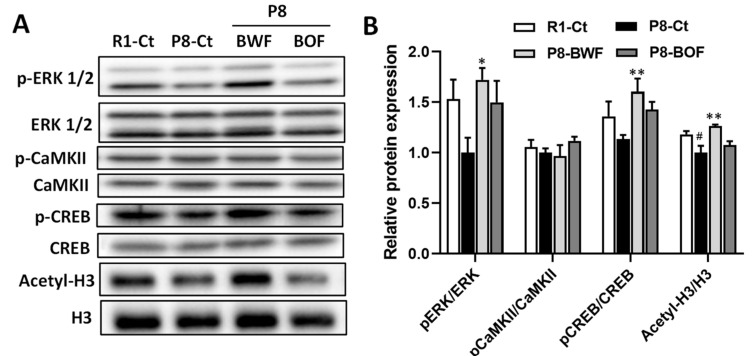
Effects of oral administration of buckwheat flours on phosphorylation of ERK, CaMKII, and CREB and acetylation of histone H3 in the hippocampus of SAMP8 mice. (**A**) Representative image of the Western blot. (**B**) Densitometric data are expressed as the ratio of phosphorylated proteins to total proteins, and the ratio of acetyl histone H3 to total histone H3. Each bar is expressed as the mean ± standard error of mean (*n* = 5). ^#^
*p* < 0.05, compared with R1-Ct group. * *p* < 0.05, ** *p* < 0.01 vs. the P8-Ct group. ERK, extracellular signal-regulated kinase; CaMKII, calcium/calmodulin-dependent protein kinase II; CREB, cAMP response element-binding protein.

**Figure 6 nutrients-14-02708-f006:**
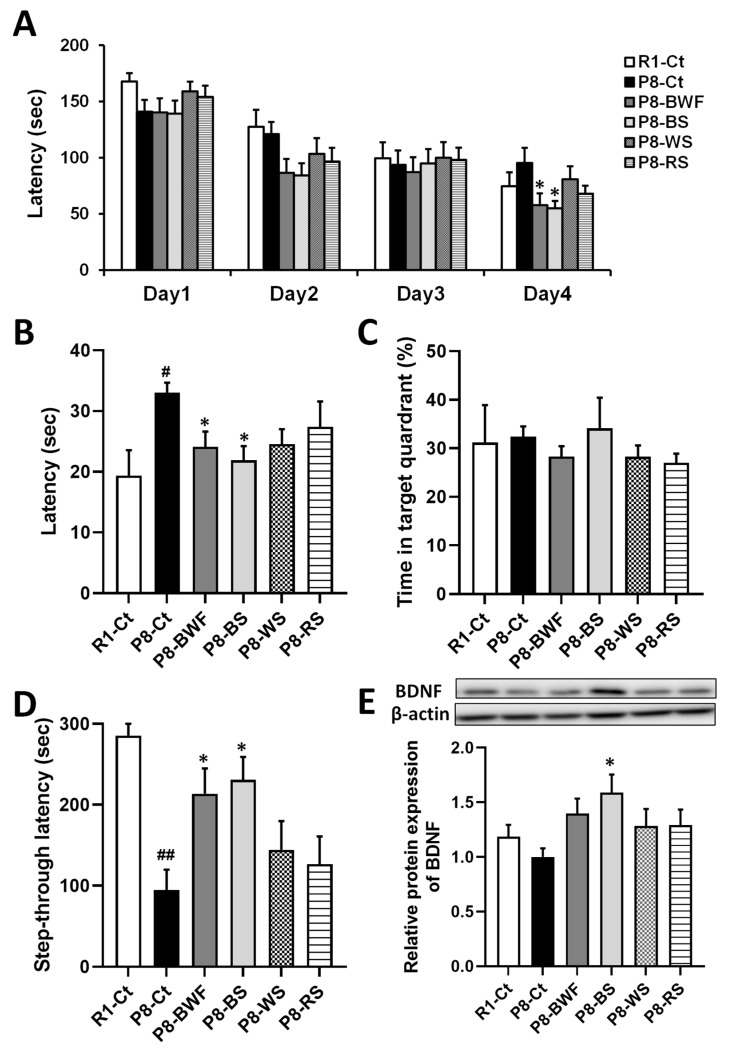
Effect of oral administration of various starch on learning and memory and hippocampal BDNF expression in SAMP8 mice. (**A**) Escape latency during training in the Barnes maze. (**B**) Escape latency during the probe test in the Barnes maze. (**C**) Time spent in target quadrant during the probe test in the Barnes maze. (**D**) Step-through latency in the passive avoidance test. (**E**) Representative image of the Western blot. β-actin was used as the internal control. Densitometric analysis of the band was expressed as relative fold compared with that in the P8-Ct group. Data are presented as the mean ± standard error of the mean (*n* = 8−10). ^#^
*p* < 0.05, ^##^
*p* < 0.01, compared with R1-Ct group. * *p* < 0.05 vs. the P8-Ct group. BS, buckwheat starch; WS, wheat starch; RS, rice starch.

**Table 1 nutrients-14-02708-t001:** Chemical composition of the flours used in this study.

	Protein (g/100 g)	Lipid (g/100 g)	Carbohydrate (g/100 g)	Dietary Fiber (g/100 g)	Rutin (mg/100 g)
BWF	13.8	3.3	68.1	4.5	6.6
BOF	33.6	7.1	44.3	16.7	13.0
TWF	12.8	3.3	69.4	3.6	808
WF	11.5	1.4	73.6	2.6	0.037
RF	6.2	0.9	82.0	0.9	ND

BWF, buckwheat whole flour; BOF, buckwheat outer flour; TWF, Tatary buckwheat whole flour; WF, wheat flour; RF, rice flour; ND, not detected.

**Table 2 nutrients-14-02708-t002:** Composition of standard and experimental diets used in the Study 1.

Ingredients (%)	Standard Diet	BWF	BOF	TWF	Rutin	WF	RF
Casein	14.00	13.31	12.32	13.36	14.00	13.43	13.69
L-cystine	0.18	0.18	0.18	0.18	0.18	0.18	0.18
Corn starch	46.57	42.65	44.44	42.55	46.57	42.34	41.97
α-cornstarch	15.50	15.50	15.50	15.50	15.50	15.50	15.50
Sucrose	10.00	10.00	10.00	10.00	10.00	10.00	10.00
Soybean oil	4.00	3.84	3.65	3.84	4.00	3.93	3.96
Cellulose	5.00	4.78	4.17	4.82	5.00	4.87	4.96
Mineral mix	3.50	3.50	3.50	3.50	3.50	3.50	3.50
Vitamin mix	1.00	1.00	1.00	1.00	1.00	1.00	1.00
Choline bitartrate	0.25	0.25	0.25	0.25	0.25	0.25	0.25
*tert*-butylhydroquinone	0.0008	0.0008	0.0008	0.0008	0.0008	0.0008	0.0008
Tested sample	0.00	5.00	5.00	5.00	0.05	5.00	5.00

BWF, buckwheat whole flour; BOF, buckwheat outer flour; TWF, Tatary buckwheat whole flour; WF, wheat flour; RF, rice flour.

**Table 3 nutrients-14-02708-t003:** Composition of standard and experimental diets used in the Study 2.

Ingredients (%)	Standard Diet	BWF	BS	WS	RS
Casein	14.00	13.31	13.97	13.99	13.99
L-cystine	0.18	0.18	0.18	0.18	0.18
Corn starch	46.57	42.65	42.64	42.61	42.60
α-cornstarch	15.50	15.50	15.50	15.50	15.50
Sucrose	10.00	10.00	10.00	10.00	10.00
Soybean oil	4.00	3.84	3.98	3.98	3.98
Cellulose	5.00	4.78	4.98	5.00	5.00
Mineral mix	3.50	3.50	3.50	3.50	3.50
Vitamin mix	1.00	1.00	1.00	1.00	1.00
Choline bitartrate	0.25	0.25	0.25	0.25	0.25
*tert*-butylhydroquinone	0.0008	0.0008	0.0008	0.0008	0.0008
Tested sample	0.00	5.00	4.00	4.00	4.00

BWF, buckwheat whole flour; BS, buckwheat starch; WS, wheat starch; RS, rice starch.

## Data Availability

The data are available from the corresponding author upon reasonable request.

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
