# Peer review of "Buckwheat Flour and Its Starch Prevent Age-Related Cognitive Decline by Increasing Hippocampal BDNF Production in Senescence-Accelerated Mouse Prone 8 Mice"

_nutrients, 2022, doi:10.3390/nu14132708_

Round 1

Reviewer 1 Report

In this study, authors showed that long-term administration of BWF attenuated age-related cognitive decline in SAMP8 mice.

Results showed that BWF treatment increased hippocampal BDNF levels, ERK/CREB signaling pathway was involved. BWF treatment increased NeuN expression level in the hippocampus.

An increase in histone H3 acetylation in the hippocampus was observed in the P8-BWF group. Study was well designed and results are interesting. I only have a few concerns as below:

1. The method of Barnes maze is not clear. How long was mouse trained for each trial? How was the probe trial was conducted? And where is the result for the probe trial? The results of Barnes maze only showed the escape latency, please also specify the total time of mice were allowed to explore before they get into the escape hole.

2. I noticed that, during the whole experiment, mouse was housed singly, why? There is study found that single housed mouse would display anxiety behavior, which might affect the learning and memory results.

3. Author mentioned that food consuming and body weight were measured, was there any difference? In the method part (2.3.1 and 2.3.2), the definition for the groups is not clear. What is the “control”, “the control group” and “sample group”?

4. Authors used female 18 weeks old SAMP8 mice, will these SAMP8 mice display learning and memory dysfunction at this age when compared with SAM resistant1 mice?  At result figure 1, there is no difference for the escape latency of Barnes maze between SAMP8 and SAM resistant 1-mice. And why choose to use female mice? Should the menstrual period be a problem for the behavior study?

5. In abstract, buckwheat was pointed out as important gluten-free crop, but rice is also gluten free, it doesn’t feel appropriate to point this out here. Gluten-free is not the causal in this study.

6. Figure 6, E: the title of Y-axis needs a little correction.

7. If possible, it would be much better if authors could supply a high power image for BDNF immunofluorescence.

Author Response

We appreciate the valuable comments by the reviewers and the editor. We have carefully revised the manuscript and now believe its quality to be substantially improved. Point-by-point responses to each comment can be found in the attachement.

Reviewer 2 Report

This is an interesting study on the effect of Buckwheat flour on cognition and various measures in the brain. It is well written and presented, however there are some details missing and other small changes I would suggest to improve the manuscript overall.

Abstract

1. For the sentence on line 19-21 I don't think you need to repeat all of the diet groups here as you have already listed them all in the last sentence. Maybe better to say "Mice fed with BWF showed improved performance compared to a control diet, while no improvement was seen following other diets" or similar.

Methods

2. The Animal treatment sections need some more details. How many mice were used/group? Was there loss of animals during the ageing process? Were half of each group collected for Western Blot (frozen in liquid nitrogen) and half for IHC (parrafin embedded- were these fixed in paraformaldehyde?) in Study 1? You should also state somewhere that microbiota and brain measures were only conducted in some diet groups, and explain why.

3. It would be good to include details about the difference between the diets used in the 2 studies somewhere in the methods, maybe section 2.3.2. This could actually be similar to what you already have at the end of the Introduction. Also in this section, what is the reason for using different ages in the 2 studies?

4. A key for each of the abbreviations for each of the diets like you have in table 1 and the figure legends might be useful for table 2 and 3 also, just to make it easier than going back to the initial definition in the methods.

5. For the Barnes maze methods, can you define "stay-in-the-hole time? Does this refer to the escape box? Were trials ended when the mouse entered the escape box? What was the maximum amount of time for each trial? I see that the reference includes extra details but I think it would be useful to include these ones here as well briefly.

6. In the Passive avoidance test methods, is Muromachi Kikai a company or a reference? I believe more information is required either way.

Results

7. Section 3.1: The 2nd two sentences are not really needed in the results section. The first sentence confirming which data is being presented is fine, although some mention of Study 1 would make it consistent with the methods and easier.

8. Can you also present your stats results throughout all section of the text (p values and possibly degrees of freedom etc)?

9. The results for the behavioural data definitely needs a lot more detail to explain and justify your conclusions. Without this I'm not sure your conclusion that the BWF diet improves cognition is actually justified throughout the rest of the paper.

Maybe start with any differences or lack of compared to the SAMR1 controls, i.e. is there actually impairment in the SAMP8 mice on the control diet that could be improved by the other diets? You seem to have followed this format for some of the brain measures, this presentation of results should be consistent throughout the manuscript.

Please explain whether the training results are from one or more days of training or across all training days? Was each day analysed separately or with a 2 way ANOVA with days included as a repeated measure? There are several significant differences on the graph that you don't explain or mention in the text at all.

When you talk about "time in the quadrant where the goals had been placed" does this refer to the probe trial? This terminology is not really consistent with previous description of the Barnes maze.

10. Make sure you specify which control group you are talking about in each section, the SAMP8 on the control diet, or the SAMR1 control mice?

11. Instead of saying "The passive avoidance test showed that BWF treatment increased the latency compared to that of the control, although no significant differences were found" maybe say ". . . BWF treatment appeared to increase the latency . . ." Can you present the stats to show if there were any trends towards significance for this observation?

12. On figure 1A, the squares on the in figure legend should be larger so we can actually see the pattern the same as on the graph columns.

13. Be consistent with the group names throughout the results and the rest of the manuscript, e.g. in section 3.2 you have just said BWF sometimes without the mouse strain, but then use P8-BOF below for the different diet group but still the same strain. You also start abbreviating the strain names in this section instead of earlier.

14. In figure 2G, Was the P8-Ct group significantly lower than R1-Ct group? It looks like both treatments restored levels to the same as R1-Ct but n may have been too low for statistical significance? This is where presenting your statistical results would be useful.

15. Section 3.4: Don't use the abbreviation in the title. Keep a consistent format with the Study 1 results sections where you have said buckwheat flour in full. State specifically that these results are from Study 2.

Discussion

16. I think the paragraph on microbiota should focus first on the increases that were seen in the BWF group, and how these could be related to the behavioural and brain changes seen.

Then a smaller amount on the changes seen in the BOF group and the fact this didn't align with other measures. The sentence "Thus, the cognitive 332 performance of SAMP8 mice may not be affected by changes in the gut microbiota caused 333 by BOF administration" seems to come out of nowhere, and needs a reminder that BOF has no effect on cognition or other brain measures, despite these differences in microbiome.

17. I find lines 334-348 (last part of the microbiome paragraph) hard to follow and understand how it relates to the current findings of your study.

Author Response

(The authors gave the same response as above.)
